# m6Aminer: Predicting the m6Am Sites on mRNA by Fusing Multiple Sequence-Derived Features into a CatBoost-Based Classifier

**DOI:** 10.3390/ijms24097878

**Published:** 2023-04-26

**Authors:** Ze Liu, Pengfei Lan, Ting Liu, Xudong Liu, Tao Liu

**Affiliations:** 1College of Water Resources and Architectural Engineering, Northwest A&F University, Xianyang 712100, China; 2Department of Mechanical Engineering, Faculty of Engineering, The University of Hong Kong, Hong Kong 999077, China; 3College of Control Science and Engineering, Zhejiang University, Hangzhou 310027, China; 4Key Laboratory of Agricultural Soil and Water Engineering in Arid and Semiarid Areas, Ministry of Education, Northwest A&F University, Xianyang 712100, China

**Keywords:** machine learning, m6Am, CatBoost, cross-validation, model optimization

## Abstract

As one of the most important post-transcriptional modifications, m6Am plays a fairly important role in conferring mRNA stability and in the progression of cancers. The accurate identification of the m6Am sites is critical for explaining its biological significance and developing its application in the medical field. However, conventional experimental approaches are time-consuming and expensive, making them unsuitable for the large-scale identification of the m6Am sites. To address this challenge, we exploit a CatBoost-based method, m6Aminer, to identify the m6Am sites on mRNA. For feature extraction, nine different feature-encoding schemes (pseudo electron–ion interaction potential, hash decimal conversion method, dinucleotide binary encoding, nucleotide chemical properties, pseudo k-tuple composition, dinucleotide numerical mapping, K monomeric units, series correlation pseudo trinucleotide composition, and K-spaced nucleotide pair frequency) were utilized to form the initial feature space. To obtain the optimized feature subset, the ExtraTreesClassifier algorithm was adopted to perform feature importance ranking, and the top 300 features were selected as the optimal feature subset. With different performance assessment methods, 10-fold cross-validation and independent test, m6Aminer achieved average AUC of 0.913 and 0.754, demonstrating a competitive performance with the state-of-the-art models m6AmPred (0.905 and 0.735) and DLm6Am (0.897 and 0.730). The prediction model developed in this study can be used to identify the m6Am sites in the whole transcriptome, laying a foundation for the functional research of m6Am.

## 1. Introduction

In recent years, post-transcriptional modifications of RNA have become an important focus of biological research. Since the discovery of the first RNA modification [1], more than 170 post-transcriptional modifications have been discovered on almost all types of RNA [2], including mRNA, tRNA, rRNA, and so on [3]. RNA modifications have demonstrated a high specificity and efficiency in regulating biological functions, such as their involvement in RNA translation [4], embryonic stem cell development [5], cancer cell survival, and migration [6]. Moreover, studies have shown that mutations in many RNA-modifying enzymes are involved in the development of human diseases, such as motor neuron disorders and autosomal-recessive intellectual disability (ARID) [7].

Among the numerous RNA modifications, a reversible modification called N6,2′-O-dimethyladenosine (m6Am) exists on mRNA and lncRNA in higher eukaryotes [8,9]. Since the first nucleotide after the cap is adenosine, it can be methylated on the 2′-hydroxyl group and further methylated at the N6 position to generate the m6Am site [10]. Therefore, the m6Am site is precisely located at the first transcribed nucleotide and is adjacent to the cap structure at the 5′-UTR of the mRNA [11]. In 1975, m6Am was first discovered in human, mouse, and adenovirus mRNAs [12]. Since then, studies have shown that a total of 50% to 80% of adenosine-initiated mammalian mRNA is modified by m6Am [11], which is catalyzed by an RNAPII-interacting enzyme phosphorylated C-terminal domain-interacting factor 1 (PCIF1) and could be removed by the m6A demethylase fat-mass and obesity-associated gene (FTO) [13,14]. Therefore, the enrichment level of m6Am in vivo is dynamically regulated by FTO and PCIF1. Due to the lack of technical means to accurately locate m6Am, there is little knowledge about the biological function of m6Am. However, recent studies have gradually revealed the importance of m6Am in regulating biological functions, such as conferring mRNA stability in mammalian cells through the resistance of the m6Am transcripts to the mRNA-unpacking enzyme-decapping protein 2 (DCP2) [15] and the potential adverse effects in mRNA for colorectal cancer therapy due to elevated m6Am levels [16]. To further investigate the biological function of m6Am, especially for the formation mechanism of related diseases or the abnormal phenomena of organisms, it is very necessary to accurately identify the m6Am sites on mRNA.

The development of transcriptome-wide sequencing technologies for various RNA modifications, including MeRIP-seq, miClIP-seq, and m6ACE-seq, has greatly advanced the field of epitranscriptomics [17]. Among these transcriptome-wide sequencing techniques, the most widely used for mapping m6Am is m6A-seq [18,19], which relies on the m6Am antibodies. However, it is difficult to discriminate the m6Am from 5′-UTR m6A [20], which seriously affects the sensitivity and precision of the m6Am detection. Although some studies have increased the confidence of m6Am via PCIF1 knockdown [21], this indirect method is not suitable for the epitranscriptome analysis of human tissues and biological samples. To address this series of problems, Sun et al. developed a method called m6Am-seq to map the m6Am in the human transcriptome [20]. This method selectively removes m6Am while maintaining the integrity of m6A through an in vitro demethylation reaction. When the identified substance cannot be eliminated by m6Am-seq, it is proven that the identified substance is m6A or otherwise m6Am. Therefore, this method can be used to distinguish m6Am from m6A at 5′-UTR.

However, the high cost, long experimental time, and low specificity of the antibodies used have certain limitations in identifying m6Am on RNA. Therefore, it is critical to develop a faster, more sensitive, and more affordable method to identify m6Am on RNA. With the increasing application of machine learning in the field of bioinformatics, many studies have developed machine-learning-based models to identify modifications on RNA or DNA, such as m7GPredictor [22], m5UPred [23], and Gene2vec [24], for the simultaneous recognition of multiple RNA modifications. In 2021, Jiang et al. proposed an XGBoost-based model called m6AmPred [25] to predict the m6Am sites. However, they did not remove redundant sequences in the benchmark dataset, which may led to the weak generalization ability of the model. In 2022, Luo et al. proposed a new predictor, named DLm6Am [26], that combines three feature extraction schemes, such as one-hot, nucleotide chemical property (NCP), and nucleotide density (ND), with a deep-learning-based classifier which demonstrates better performance than m6AmPred. However, deep learning pays attention to the automatic learning features of sequences and ignores the sequences’ physical, chemical, and structural properties. Although their work has contributed to predicting the m6Am sites, there is still room for improvement in the m6Am prediction.

Therefore, a CatBoost-based model was developed in this study (Figure 1). Sequence features were extracted by utilizing nine reliable feature extraction methods, including pseudo electron–ion interaction potential, hash binary conversion method, dinucleotide binary encoding, NCP, pseudo k-tuple composition, dinucleotide numerical mapping, K monomeric units, series correlation pseudo trinucleotide composition, and K-spaced nucleotide pair frequency. For feature optimization, the feature importance scores were calculated using the ExtraTreesClassifier algorithm [27], and the top 300 features were selected as the optimal feature subset, which was used to train the CatBoost-based model. Comprehensive comparison results show that our proposed model achieves competitive performance compared with the state-of-the-art predictors m6AmPred and DLm6Am.

## 2. Results

### 2.1. Model Performance with Different Machine Learning Algorithms and Feature Subsets

To find the appropriate machine learning algorithm, five different classifiers, AdaBoost, random forest (RF), K nearest neighbors (KNN), support vector machine (SVM), and CatBoost, were implemented and trained on the 10 sub-training datasets. For feature extraction, 1120 features were extracted using the above-mentioned 9 feature encoding schemes on the 10 sub-training datasets. For model training, 10-fold cross-validation was used on each sub-training dataset.

As shown in Table 1, the CatBoost-based model achieved the best performance, with an average ACC of 0.834 ± 0.003, an AUC of 0.912 ± 0.002, an Sn of 0.805 ± 0.004, an Sp of 0.863 ± 0.004, an F1 of 0.829 ± 0.003, and an MCC of 0.669 ± 0.006. Compared with the other four classifiers, the CatBoost-based model successfully increased the ACC by over 0.7%, the AUC by over 1%, the Sn by over 0.7%, the F1 by over 0.8%, and the MCC by over 1.2%. Although the Sp value of SVM was higher than that of CatBoost, the other metrics were significantly lower than the CatBoost metrics (Figure 2). Simultaneously, for the RF-based model, the standard deviations (SDs) of six metrics except for Sp were the lowest, indicating that RF has the least volatility on different data sets. However, the six metrics of RF were significantly lower than those of CatBoost. Therefore, the m6Aminer utilized the CatBoost algorithm to construct the model.

To compare its performance with different feature subsets, the CatBoost-based model was trained on the 10 sub-training datasets with different feature subsets. The hyperparameters of the CatBoost classifier were set to their default parameters. Using different feature-encoding schemes, the mean AUC values of our models were all above 0.870 (Table 2). Among the models, PseEIIP achieved the highest AUC of 0.910 ± 0.002 on the 10 sub-training datasets when adopting a single feature extraction method. However, the other schemes also showed decent performance on the 10 sub-training datasets. As shown in Table 2, the performance achieved when combining multiple feature encoding schemes for our model was better than when a single method was utilized.

### 2.2. Feature Ranking and Selection

The ExtraTreesClassifier algorithm is an ensemble classifier utilized for feature ranking. The importance scores of features also can be obtained to analyze the performance of the nine feature extraction methods (Appendix A). It can be seen that DNM, K-mer, Ksnpf, PseEIIP, and SCPseTNC showed a high correlation with the classification results (Figure 3A), indicating that these features play a fairly important role in this classification task. The importance of the top 20 important features (Figure 3B) also shows that the features extracted via DNM, K-mer, Ksnpf, PseEIIP, and SCPseTNC have a greater impact on model performance. It can be distinctly seen that the proportions of K-mer, PseEIIP, and SCPseTNC in the top 20 features are 30% (6/20), 25% (5/20), and 20% (4/20), respectively. These results confirm that there is a strong correlation between the electronic properties and the physical and chemical properties of the corresponding nucleotide and the accurate recognition of the m6Am sites.

The equidistance method was then used to select the key features. The top 50 features were used for the first iteration, and the second top 50 ranked features were then added to the feature subset in subsequent iterations, such as the top 50 features for the first iteration and the top 100 features for the second iteration, until the 1120 features were utilized to train the model (using 10-fold cross-validation on the 10 sub-training datasets). As shown in Figure 3C, our model obtained the best performance using the top 300 important features for feature encoding, and our model received an average AUC of 0.912 ± 0.002, ACC of 0.832 ± 0.002, Sn of 0.804 ± 0.003, Sp of 0.861 ± 0.005, F1 of 0.827 ± 0.002, MCC of 0.666 ± 0.005, respectively (the detailed results are listed in Appendix A).

### 2.3. Model Optimization

After feature selection, GridSearchCV was utilized to find the best hyperparameters, e.g., “iterations”, “depth”, and “learning_rate”. In this study, “iterations” was selected from {250, 100, 500, 1000, 1500, 2000}, “depth” was selected from 1 to 10, and “learning_rate” was selected from {0.001, 0.01, 0.03, 0.1, 0.2, 0.3}. The important hyperparameters of the m6Aminer are listed in Table 3 (The detailed hyperparameters are listed in Appendix A).

### 2.4. Comparison with the State-of-the-Art Predictors

To further evaluate the performance of our model, m6Aminer was compared with the state-of-the-art predictors m6AmPred [25] and DLm6Am [26]. In m6AmPred, PseEIIP was used for feature extraction, and extreme gradient boosting with the dart algorithm (XgbDart) was used for model training. The grid research method was utilized, and m6AmPred received the best gamma = 1.05, learning_rate = 0.1, and eta = 0.03. Other parameter settings refer to the description by Jiang et al. [25]. In DLm6Am, one-hot, ND, and NCP were utilized for extraction, and the deep learning algorithm was used to train the model. The detail parameter settings were set as in Luo et al. [26]. These two models were both trained and optimized on our dataset with the same fold to ensure the fairness of comparison.

As shown in Table 4, our model achieved an average ACC of 0.834 ± 0.003, AUC of 0.913 ± 0.002, Sn of 0.806 ± 0.003, Sp of 0.861 ± 0.005, F1 of 0.829 ± 0.003, MCC of 0.668 ± 0.006 on the 10 sub-training datasets using 10-fold cross-validation. The six metrics of our model were higher than those of the other two models, and m6Aminer successfully increased the average ACC by over 0.7%, AUC by over 0.8%, Sn by over 0.3%, Sp by over 0.3%, F1 by over 0.7%, and MCC by over 1.3%, demonstrating that our predictor achieved better performance in identifying m6Am than the state-of-the-art predictors m6AmPred and DLm6Am. Simultaneously, the SD of the average AUC values for m6Aminer was 0.002, showing that the stability of our model is comparable with the state-of-the-art models.

As shown in Table 5, our model received an average AUC of 0.754 ± 0.005, ACC of 0.647 ± 0.009, Sn of 0.874 ± 0.006, Sp of 0.420 ± 0.024, F1 of 0.713 ± 0.004, and MCC of 0.331 ± 0.015 on the independent testing dataset, which increased the average AUC by over 1.9%, ACC by over 0.5%, Sp by over 1.1%, F1 by over 0.3%, and MCC by over 0.9% (the details of the independent testing results are listed in Appendix A). Except for Sn, the other measurements were higher than those of m6AmPred and DLm6Am. Concurrently, the SDs of the six metrics for m6Aminer were lower than that of the other two models, proving that m6Aminer has the best stability on different datasets. This best stability indicates that our model achieves stronger competitiveness in terms of its generalization ability when compared to the other two models with a completely new dataset. To more intuitively compare the effectiveness of the three predictors, the ROC curves and precision–recall curves are also shown in Figure 4 and Figure 5, respectively.

### 2.5. Web Server

To save the experimental cost of m6Am identification, simplify the experimental process, and reduce the experimental time, a web server based on the flask framework was designed to predict the m6Am sites. It can be accessed at www.m6aminer.cn (accessed on 3 April 2023). It allows users to upload the RNA sequences for identification by submitting the fasta format sequences through the text box. As the length of the RNA sequence used for the training model was 41, the RNA sequence input by users should also be 41nt. The online tool can be used to predict the m6Am sites in real time. The Tencent cloud server will automatically process the data and then feed the predicted results back to the result webpage for the user’s reference. The specific page of the webpage is shown in Figure 6. The source code for m6Aminer is also provided in the download section, which allows users to run the model locally.

## 3. Discussions

Accurately predicting the existence of m6Am on RNA is particularly important for revealing the biological function of m6Am more deeply. In this study, we developed a new computational model, m6Aminer, to improve the prediction of m6Am. The model innovatively utilized nine feature schemes (PseEIIP, DBE, Hash, NCP, PseKNC, DNM, K-mer, SCPseTNC, and Ksnpf), and selected the top 300 features using the ExtraTreesClassifier algorithm. It was found that the features extracted by K-mer, PseEIIP, and SCPseTNC accounted for the majority of the top 20 features, suggesting that the electronic, physical, and chemical properties of nucleotides have an important impact on the classification task. To the best of our knowledge, the currently available models for predicting m6Am using machine learning are m6AmPred and DLm6Am. Compared to the above models, m6Aminer excavated more important features related to the m6Am site. Additionally, our model does not consume GPU resources and requires little training time. According to the comprehensive comparison results of this study, it can be seen that our predictor achieves competitive performance when compared with m6AmPred and DLm6Am. Therefore, our method can be used as an efficient tool for identifying m6Am or can at least become an auxiliary means of identifying m6Am in the future.

However, our method has some limitations. Firstly, the limited experimental data have a great impact on the generalization ability of the model. In the future, more sequences containing the m6Am sites will be collected to enhance the recognition accuracy for the m6Am sites. Furthermore, this study only used sequence features. In future studies, RNA secondary structure features or genomic features could also be used to extract more useful features for identifying the m6Am sites. Finally, some more advanced algorithms should be used to optimize our model, such as transfer learning.

## 4. Materials and Methods

### 4.1. Benchmark Dataset

In this study, benchmark datasets were downloaded from http://47.94.248.117/DLm6Am (accessed on 3 April 2023) and https://www.xjtlu.edu.cn/biologicalsciences/m6am (accessed on 3 April 2023). To remove sequence redundancy, the benchmark datasets were screened with CD-HIT, using the most rigorous threshold of 0.8 [28]. A total of 3700 positive samples and 37,000 negative samples were assigned to the training dataset after the above operation, while the independent testing dataset consisted of 320 positive samples and 320 negative samples. To construct an approximate 1:1 positive-to-negative ratio dataset, 37,000 negative samples were split into 10 negative datasets. Ten sub-training datasets were generated by combining each of these then negative datasets with a positive dataset. Their prediction performances were averaged during the evaluation to reduce batch variance [25]. Furthermore, the 10 sub-training datasets were used for feature selection and model construction, while the independent testing datasets only participated in the independent test.

### 4.2. Feature Extraction

#### 4.2.1. Pseudo Electron–Ion Interaction Potential (PseEIIP)

A feature extraction method using the electron–ion interaction potential (EIIP) values of nucleotides was pioneered by Nair and Sreenadhan [29]. The EIIP value is calculated from the energy of the delocalized electrons in amino acids or nucleotides. This method was initially used to map exons and has been widely used to build models to identify some modifications, such as m5C [30,31,32].

In PseEIIP, the extracted features are divided into two parts: the distribution of free-electron energy along the RNA sequence and the product of three consecutive nucleotides’ free-electron energies and frequencies. In the portion of the free-electron energy distributed along the RNA sequence, the nucleotide of each RNA sequence is encoded as a numerical value representing its electron–ion interaction potential. The values corresponding to each nucleotide are shown in Table 6. In the second part, the three consecutive nucleotides’ free-electron energies and their frequencies of occurrence are calculated separately. The free-electron energy of any three continuous nucleotides is compiled by that of three individual ones, and the frequency is the current tri-consecutive nucleotide’s total number divided by those occurrence times. The formula is as follows:(1)$$EIIP_{xyz}=EIIP_{x}+EIIP_{y}+EIIP_{z}$$
(2)$$f_{xyz}=\frac{N_{xyz}}{L-2}$$

Among these variables, $N_{xyz}$ represents the number of occurrences of three consecutive nucleotides in a sequence, and *L* represents the sequence length. Therefore, for any given RNA sequence, a feature vector of *L* + 64 dimensions can be obtained by using the PseEIIP method. In this study, the sequence length of each sample was 41nt, and a total of 105 features were extracted from an RNA sequence. The feature vectors are as follows:(3)$$PseEIIP=[EIIP_{AAA}f_{AAA},EIIP_{AAC}f_{AAC},\cdots \cdots ,EIIP_{TTT}f_{TTT}]$$

#### 4.2.2. Hash Decimal Conversion Method (Hash)

Hash [33] is a method of converting quaternary numbers into decimals, which represent the four bases in RNA sequences (A, C, G, and U). For an RNA sequence, the contiguous two bases can be considered a two-digit-quaternary number. Therefore, the sample of 41nt can be turned into 40 two-digit quaternary numbers. The transformation criteria for four bases when using hash are ‘A’—0, ‘G’—1, ‘C’—2, and ‘U’—3. Subsequently, each two-digit quaternary number can be converted into a decimal via the formula:(4)$$s=\sum_{i=1}^{k}4^{k-i}*h_{i}$$

Supposing that the length of each sample is *m* and the length of the selected bases is *k*, the number of the final decimal number obtained is *m* − *k* + 1. For the dataset used in the study, a 40-dimension feature vector was generated for each sample.

#### 4.2.3. Dinucleotide Binary Encoding (DBE)

DBE [34] is a method of encapsulating the positional information of the dinucleotide at each position in the sequence. It has been extensively utilized to construct the predictor, such as M6AMRFS [35] and iMRM [36]. Each dinucleotide can be encoded into a four-dimensional 0/1 vector with DBE. For example, AA may be encoded as (0,0,0,0), AT is encoded as (0,0,0,1), AC is encoded as (0,0,1,0), and GG is encoded as (1,1,1,1). Consequently, a sequence of 41nt with DBE could be encoded as a 160-dimensional vector.

#### 4.2.4. Nucleotide Chemical Property (NCP)

NCP was first proposed by Bari et al. to predict the splice sites in pre-mRNA [37]. Since the four nucleotides in the RNA sequence may provide different functional structures according to their chemical structures, three coordinate values were used to represent the different chemical properties of the four nucleotides. Firstly, both purines (A and G) and pyrimidines (C and U) have rings; purines have two rings, and pyrimidines have only one ring. Therefore, we used the x-coordinate information to distinguish the ring structures of purines and pyrimidines. Secondly, through the y-coordinate information, we judged the existence of amino groups (A and C) or keto groups (G and U). Finally, the z-coordinate information was determined via strong hydrogen bonds (C and G) and weak hydrogen bonds (A and U). According to the different chemical properties of the above four nucleotides, the encoding of a nucleotide in an RNA sequence satisfies the following formula:(5)$$x_{i}=\left\{\begin{array}{cc} 1 & if\;s_{i}\in \left\{A,G\right\}\\ 0 & if\;s_{i}\in \left\{C,U\right\} \end{array}\right.\;y_{i}=\left\{\begin{array}{cc} 1 & if\;s_{i}\in \left\{A,C\right\}\\ 0 & if\;s_{i}\in \left\{G,U\right\} \end{array}\right.\;z_{i}=\left\{\begin{array}{cc} 1 & if\;s_{i}\in \left\{A,U\right\}\\ 0 & if\;s_{i}\in \left\{C,G\right\} \end{array}\right.$$

In detail, the four nucleotides A, C, G, and U were encoded as vectors (1,1,1), (0,1,0), (1,0,0), and (0,0,1), respectively. Therefore, each nucleotide in an RNA sequence was present as three elements, and a total of 123 features were extracted for each sample in the study.

#### 4.2.5. Pseudo k-Tuple Composition (PseKNC)

PseKNC has been successfully applied to the identification of RNA or DNA modification by forming the physicochemical properties of oligonucleotides [38], such as MSLP [39], iterb-PPse [40], and mRNALocater [41]. Each sample can be expressed utilizing the following formula:(6)$$D={\left[d_{1}d_{2}\ldots d_{4^{k}}d_{4^{k}}{}_{+1}\ldots d_{4^{k}}{}_{+\lambda }\right]}^{T}$$
where:(7)$$d_{n}=\left\{\begin{array}{c} \frac{f_{n}}{\sum_{i=1}^{4^{k}}f_{i}+\omega \sum_{j=1}^{\lambda }\theta_{j}}(1\leq n\leq 4^{k})\\ \frac{\omega \theta_{n-4^{k}}}{\sum_{i=1}^{4^{k}}f_{i}+\omega \sum_{j=1}^{\lambda }\theta_{j}}(4^{k}\leq n\leq 4^{k}+\lambda ) \end{array}\right.$$
(8)$$\theta_{j}=\frac{1}{L-j-1}\sum_{i=1}^{L-j-1}\Phi (R_{i}R_{i+1},R_{i+j}R_{i+j+1})\;\left(1\leq j\leq \lambda ;\lambda <L\right)$$
(9)$$\Phi (R_{i}R_{i+1},R_{i+j}R_{i+j+1})=\frac{1}{\mu }\sum_{u=1}^{\mu }[P_{u}(R_{i}R_{i+1})-P_{u}(R_{i+j}R_{i+j+1}){]}^{2}$$
where $f_{n}$ is the normalized frequency of 16 dinucleotides in the RNA sequences, $\theta_{j}$ is the *j*-tier sequence correlation factor, and λ is the top count level of correlation in the samples.

#### 4.2.6. Dinucleotide Numerical Mapping (DNM)

DNM is a feature extraction method proposed by Zu et al. which can be used to obtain a 48-dimensional feature vector based on the average, expectation, and variance of the dinucleotide [42]. For each sequence, 16 dinucleotides can be extracted by using the four bases (A, G, C, and U). For the AA dinucleotide, the detailed solution procedure is shown as follows:(10)$$\begin{array}{l} f_{AA}(t_{i})=1,t_{i}=AA\\ f_{AA}(t_{i})=0,t_{i}\neq AA\\ i=1,2,\cdots \cdots ,m-1 \end{array}$$
where *m* is the length of the RNA sequence and was set to 41 in this study, and $f_{AA}(t_{i})$ is the criterion for judging whether the dinucleotide is AA or not. For a sequence, if the two adjacent bases are AA, $f_{AA}(t_{i})=1$; otherwise $f_{AA}(t_{i})=0$. The statistics of average, expectation, and variance are named $n_{AA}$, $u_{AA}$ and $D_{2}^{AA}$ and can be calculated using the following formula:(11)$$n_{AA}=\sum_{i=1}^{m-1}f_{AA}(t_{i})\;u_{AA}=\sum_{i=1}^{m-1}i*\frac{f_{AA}(t_{i})}{n_{AA}}\;D_{2}^{AA}=\sum_{i=1}^{m-1}\frac{{\left(i-u_{AA}\right)}^{2}f_{AA}(t_{i})}{n_{AA}(m-1)}$$

The average, expectation, and variance of the other 15 dinucleotides (AC, AG, AT, …, TG, TT) are also obtained through the above calculation method.

#### 4.2.7. K Monomeric Units (K-mer)

As a simple and effective feature extraction method, K-mer has been widely used in a variety of prediction models [43,44]. The feature extraction principle of K-mer is to count the number of occurrences of k consecutive nucleotides in the RNA sequence. If k is set to three, the number of occurrences of 64 consecutive trinucleotides is calculated, including AAA, AAT, …, GGG. To calculate the K-mer of each sample, the counting range of a sequence of length *L* is obtained from the initial nucleotide until the *L* − *k* + 1 th nucleotide. In this study, the values of k were set to 2, 3, and 4, respectively. Therefore, a total of 336 sequence features were extracted via K-mer.

#### 4.2.8. Series Correlation Pseudo Trinucleotide Composition (SCPseTNC)

SCPseTNC was proposed in a previous study by Chen et al. [45]. In the sequence-related pseudo-trinucleotide composition feature extraction method, the expression of RNA-seq feature vectors combines 12 built-in trinucleotide physicochemical indicators.

For a given RNA sequence, the feature vector extracted by SCPseTNC is defined as follows:(12)$$Vec={\left[d_{1}d_{2}\cdots d_{64}\cdots d_{64+\lambda }\cdots d_{64+\lambda \wedge }\right]}^{T}$$
where:(13)$$d_{k}=\left\{\begin{array}{cc} \frac{f_{k}}{\sum_{i=1}^{64}f_{i}+\omega \sum_{j=1}^{\lambda }\theta_{j}} & \left(1\leq k\leq 64\right)\\ \frac{\omega \theta_{k-64}}{\sum_{i=1}^{64}f_{i}+\omega \sum_{j=1}^{\lambda \Lambda }\theta_{j}} & \left(65\leq k\leq 64+\lambda \Lambda \right) \end{array}\right.$$

$f_{k}$ is the normalized frequency of the occurrence of trinucleotides in each sample; λ is an integer, representing the highest count rank or layer associated along the RNA sequence; ω is a weighting factor from 0 to 1; Λ is the number of physicochemical properties; $\theta_{j}$ is called the j-layer correlation factor, which reflects the sequence order correlation between all the most adjacent trinucleotides in an RNA sequence and is defined as follows:(14)$$\begin{array}{cc} \theta_{1}=\frac{1}{L-4}\sum_{i=1}^{L-4}J_{i,i+1}^{1} & \lambda <(L-3)\\ \theta_{2}=\frac{1}{L-4}\sum_{i=1}^{L-4}J_{i,i+1}^{2} & \lambda <(L-3)\\ \cdots & \\ \theta_{\wedge }=\frac{1}{L-4}\sum_{i=1}^{L-4}J_{i,i+1}^{\wedge } & \lambda <(L-3)\\ \cdots & \\ \theta_{\lambda \Lambda -1}=\frac{1}{L-\lambda -3}\sum_{i=1}^{L-\lambda -3}J_{i,i+\lambda }^{\Lambda -1} & \lambda <(L-3)\\ \theta_{\lambda \Lambda }=\frac{1}{L-\lambda -3}\sum_{i=1}^{L-\lambda -3}J_{i,i+\lambda }^{\Lambda } & \lambda <(L-3) \end{array}$$
where:(15)$$J_{i,i+m}^{\xi }=P_{u}(R_{i}R_{i+1}R_{i+2})P_{u}(R_{i+m}R_{i+m+1}R_{i+m+2})\;\xi =1,2,\cdots ,\Lambda \;m=1,2,\cdots \lambda \;i=1,2,\cdots ,L-\lambda -3$$

*μ* represents the number of physical and chemical properties; $P_{u}(R_{i}R_{i+1}R_{i+2})$ represents the numerical value of the physical and chemical properties of the trinucleotide at a certain position. In this study, a total of 64 sequence features were extracted by using SCPseTNC.

#### 4.2.9. K-Spaced Nucleotide Pair Frequency (Ksnpf)

Ksnpf is an effective extraction method and has been successfully applied to the prediction of DNA N4-methylcytosine sites [46] and RNA 5-methylcytosine (m5C) sites [47]. The formula to calculate Ksnpf is shown as follows:(16)$$f\left(n1Gap\left(k\right)n2\right)=\frac{S\left(n1Gap\left(k\right)n2\right)}{L-k-1}$$
where *L* is the length of each sample; $n_{1}$ and $n_{2}$ are the nucleotides, one of A, G, C, and U; $Gap\left(k\right)$ represents *k* arbitrary elements in the interval; $S\left(n1Gap\;\left(k\right)\;n2\right)$ represents the number of occurrences of element pairs. In this study, the value range of *k* was set from 1 to 5. Thus, an 80-dimensional (4 × 4 × 5) vector was obtained by utilizing Ksnpf.

### 4.3. CatBoost

CatBoost is an extension framework of gradient boosting (GB), which has been successfully applied in various fields such as driving style recognition [48] and diabetes prediction [49]. It was developed by Prokhorenkova et al. in 2017 [50,51]. This algorithm can guarantee that all features can be trained using the manner of ranking promotion and effectively reduce the risk of overfitting [52]. CatBoost utilizes the following strategies to solve the problem of overfitting.

First, the supposition is that D is the dataset provided in this study:(17)$$D=(X_{i},Y_{i})$$
where *n* is the number of samples (*i* = 1,2,…,*n*), $X_{i}=(x_{i}^{1},x_{i}^{2},\ldots ,x_{i}^{m})$ is a feature vector of a sample, $x_{i}^{m}$ is the *m*th feature vector of the sample, and $Y_{i}$ is the target value of the sample. The feature value can be converted via Formula (18):(18)$${\hat{x}}^{k}=\frac{\sum_{j=1}^{n}\varphi (x_{j}^{k}=x_{i}^{k})Y_{j}+\alpha p}{\sum_{j=1}^{n}\varphi (x_{j}^{k}=x_{i}^{k})+\alpha }$$
where α is the a priori weight, φ is the indicator function, and *p* is the prior value.

### 4.4. Performance Evaluation

In this study, 10-fold cross-validation was used to evaluate the performance of m6Aminer. Every sub-training dataset was split into 10 roughly equal parts wherein 9 folds were regarded as a training dataset, and the remaining fold was considered a validation dataset for a 10-fold cross-validation. This process was repeated until all folds were selected as the validation dataset. Finally, the average of 10 validation results was used for the performance evaluation. Additionally, six widely used metrics in binary classification were used to evaluate the performance of our model, including accuracy (ACC), sensitivity (Sn), specificity (Sp), F1-score (F1), Matthew’s correlation coefficient (MCC), and the area under the ROC curve (AUC). The detailed formulas of the six metrics are shown as follows:(19)$$\begin{array}{c} Sn=\frac{TP}{TP+TN}\\ Sp=\frac{TN}{TN+FP}\\ Acc=\frac{TP+TN}{TP+TN+FP+FN}\\ F1=\frac{2\times TP}{2\times TP+FP+FN}\\ MCC=\frac{TP\times TN-FP\times FN}{\sqrt{\left(TP+FP)\times (TP+FN)\times (TN+FP)\times (TN+FN\right)}} \end{array}$$
where *TP, TN, FP,* and *FN* represent the numbers of true positive, true negative, false positive, and false negative results, respectively. ACC is the most basic evaluation metric, which describes whether the prediction of the overall results is correct. The ROC curve is drawn according to the dynamic thresholds, with the true positive rate (sensitivity) as the ordinate and the false positive rate (1-specificity) as the abscissa. The AUC is the area under the ROC curve. As a frequently used metric, the AUC can reliably eliminate the impact of sample category imbalance on the indicator results. Normally, the AUC value is between 0.5 and 1. If the AUC is equal to 1, the classifier is proven to be a perfect classifier. On the contrary, if the AUC is equal to 0.5, the classifier is a random classifier, indicating that the model does not have predictive significance. Substantially, the higher the AUC value is, the better the model performs.

## Figures and Tables

**Figure 1 ijms-24-07878-f001:**
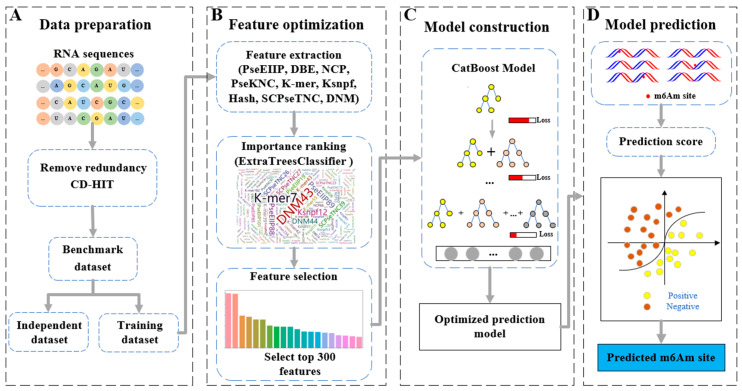
The framework of m6Aminer. (**A**). Utilize CD-HIT to remove the redundant sequences and build the benchmark datasets. (**B**). Form an initial feature space and select the top 300 features according to their importance. (**C**). Construct and optimize the CatBoost-based model. (**D**). Predict the m6Am sites using m6Aminer.

**Figure 2 ijms-24-07878-f002:**
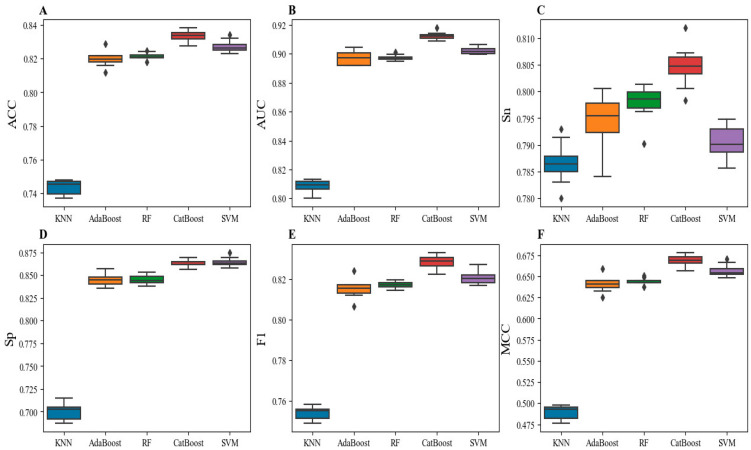
Box plots of the average metrics with different classifiers on 10 sub-training datasets. (**A**) ACC. (**B**) AUC. (**C**) Sn. (**D**) Sp. (**E**) F1. (**F**) MCC.

**Figure 3 ijms-24-07878-f003:**
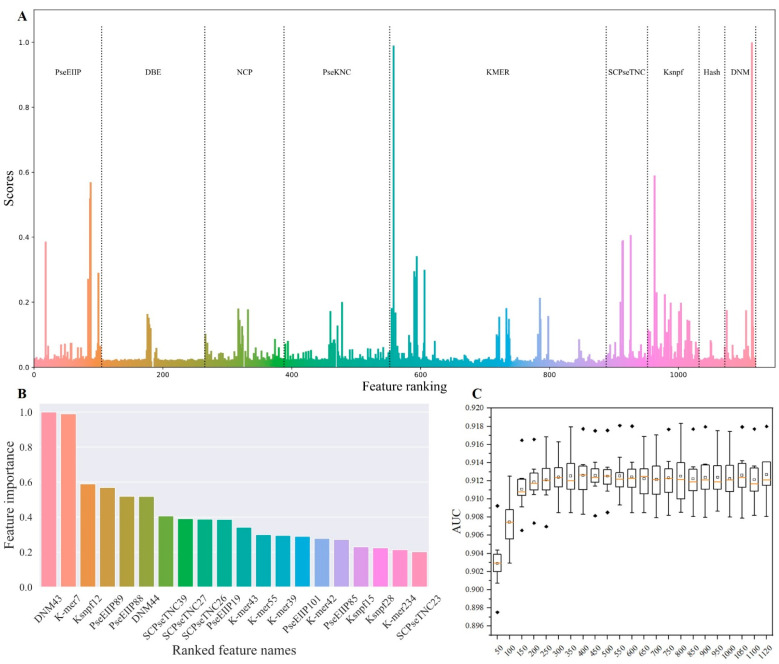
Feature ranking and selection. (**A**) The importance of 1120 features on a sub-training dataset; (**B**) The top 20 features ranked using the ExtraTreesClassifier algorithm; (**C**) The performance of the CatBoost-based model using different combinations of feature subsets on the 10 sub-training datasets.

**Figure 4 ijms-24-07878-f004:**
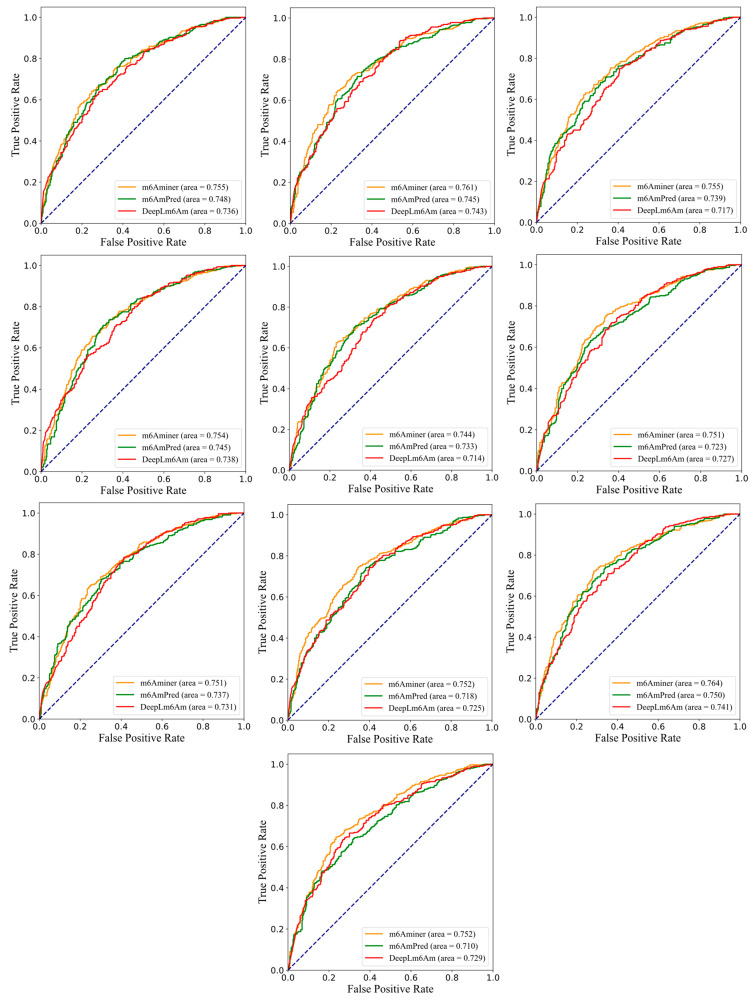
Comparison of the ROC curves for three models on the independent testing datasets.

**Figure 5 ijms-24-07878-f005:**
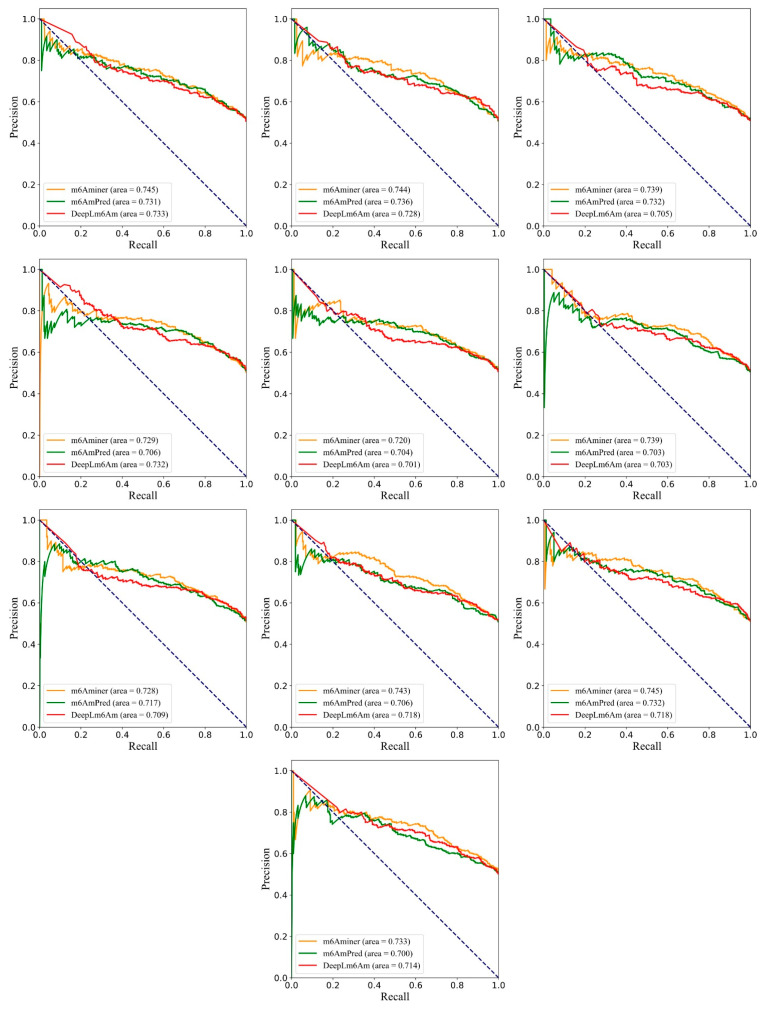
Comparison of the precision–recall curves for three models on the independent testing datasets.

**Figure 6 ijms-24-07878-f006:**
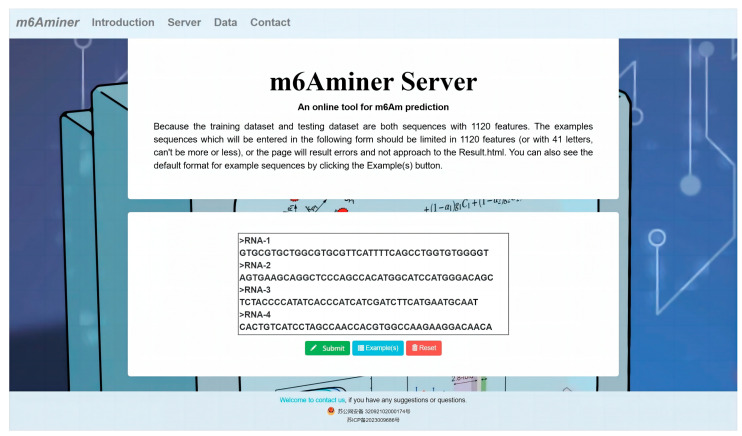
The m6Aminer web server.

**Table 1 ijms-24-07878-t001:** Model performance of different classifiers on the 10 sub-training datasets.

Classifier	ACC	AUC	Sn	Sp	F1	MCC
AdaBoost	0.820 ± 0.004	0.897 ± 0.004	0.794 ± 0.005	0.845 ± 0.006	0.815 ± 0.004	0.641 ± 0.009
RF	0.821 ± 0.002	0.897 ± 0.002	0.798 ± 0.003	0.845 ± 0.005	0.817 ± 0.002	0.644 ± 0.004
KNN	0.743 ± 0.004	0.809 ± 0.004	0.787 ± 0.004	0.700 ± 0.009	0.754 ± 0.003	0.489 ± 0.008
SVM	0.827 ± 0.003	0.902 ± 0.002	0.791 ± 0.003	0.864 ± 0.005	0.821 ± 0.003	0.657 ± 0.007
CatBoost	0.834 ± 0.003	0.912 ± 0.002	0.805 ± 0.004	0.863 ± 0.004	0.829 ± 0.003	0.669 ± 0.006

Each value in the table is represented by mean ± SD. Among them, the mean is an average of the training results of each classifier on the 10 sub-training datasets, while the SD is their standard deviation.

**Table 2 ijms-24-07878-t002:** Model performance of the CatBoost-based model using different feature subsets.

Feature	ACC	AUC	Sn	Sp	F1	MCC
PseEIIP	0.831 ± 0.003	0.910 ± 0.002	0.804 ± 0.002	0.859 ± 0.006	0.827 ± 0.003	0.664 ± 0.006
DBE	0.811 ± 0.003	0.892 ± 0.003	0.800 ± 0.002	0.822 ± 0.004	0.809 ± 0.003	0.623 ± 0.006
Hash	0.813 ± 0.003	0.891 ± 0.003	0.788 ± 0.004	0.838 ± 0.004	0.808 ± 0.003	0.627 ± 0.007
NCP	0.808 ± 0.003	0.890 ± 0.003	0.794 ± 0.002	0.822 ± 0.005	0.805 ± 0.003	0.617 ± 0.006
PseKNC	0.804 ± 0.003	0.885 ± 0.003	0.788 ± 0.004	0.820 ± 0.006	0.801 ± 0.003	0.609 ± 0.006
Kmer	0.811 ± 0.002	0.881 ± 0.002	0.776 ± 0.003	0.845 ± 0.005	0.804 ± 0.002	0.623 ± 0.005
SCPseTNC	0.808 ± 0.004	0.878 ± 0.003	0.779 ± 0.002	0.838 ± 0.006	0.802 ± 0.003	0.618 ± 0.007
DNM	0.805 ± 0.004	0.875 ± 0.003	0.773 ± 0.005	0.838 ± 0.006	0.799 ± 0.004	0.612 ± 0.007
Ksnpf	0.806 ± 0.003	0.874 ± 0.002	0.771 ± 0.004	0.842 ± 0.006	0.799 ± 0.003	0.614 ± 0.006
PseEIIP + DBE	0.832 ± 0.003	0.910 ± 0.002	0.805 ± 0.002	0.858 ± 0.005	0.827 ± 0.003	0.665 ± 0.006
PseEIIP + DBE + Hash	0.832 ± 0.002	0.910 ± 0.003	0.803 ± 0.002	0.862 ± 0.004	0.827 ± 0.002	0.666 ± 0.005
ALL	0.834 ± 0.003	0.912 ± 0.002	0.805 ± 0.004	0.863 ± 0.004	0.829 ± 0.003	0.669 ± 0.006

The mean is an average of the training results of each feature encoding scheme on the 10 sub-training datasets, while the SD is their standard deviation.

**Table 3 ijms-24-07878-t003:** The important hyperparameters of the CatBoost-based model.

Hyper-Parameters	Optimal Values
Iterations	2000
learning_rate	0.03
Depth	9
leaf_estimation_method	“Newton”
loss_function	“MultiClass”
bootstrap_type	“Bayesian”

**Table 4 ijms-24-07878-t004:** Model comparison with the state-of-the-art predictors on the 10 sub-training datasets.

Model	ACC	AUC	Sn	Sp	F1	MCC
m6Aminer	0.834 ± 0.003	0.913 ± 0.002	0.806 ± 0.003	0.861 ± 0.005	0.829 ± 0.003	0.668 ± 0.006
m6AmPred	0.826 ± 0.003	0.905 ± 0.002	0.803 ± 0.002	0.849 ± 0.006	0.822 ± 0.003	0.653 ± 0.007
DLm6Am	0.827 ± 0.002	0.897 ± 0.002	0.796 ± 0.005	0.858 ± 0.005	0.821 ± 0.002	0.655 ± 0.004

The mean is an average of the 10-fold cross-validation results of each predictor on the 10 sub-training datasets, while the SD is their standard deviation.

**Table 5 ijms-24-07878-t005:** Model comparison with the state-of-the-art models on the independent testing dataset.

Model	ACC	AUC	Sn	Sp	F1	MCC
m6Aminer	0.647 ± 0.009	0.754 ± 0.005	0.874 ± 0.006	0.420 ± 0.024	0.713 ± 0.004	0.331 ± 0.015
m6AmPred	0.623 ± 0.013	0.735 ± 0.013	0.887 ± 0.009	0.358 ± 0.024	0.702 ± 0.009	0.289 ± 0.028
DLm6Am	0.642 ± 0.014	0.730 ± 0.009	0.875 ± 0.009	0.409 ± 0.029	0.710 ± 0.008	0.322 ± 0.026

The mean is an average of the independent testing results of each predictor on the 10 sub-training datasets, while the SD is their standard deviation.

**Table 6 ijms-24-07878-t006:** EIIP values of four nucleotides.

Nucleotides	EIIP Value
A	0.1260
U	0.1335
G	0.0806
C	0.1340

## Data Availability

The dataset used in this study and the m6Aminer source code are available at https://github.com/NWAFU-LiuLab/m6Aminer (accessed on 3 April 2023).

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
