# Peer review of "m6Aminer: Predicting the m6Am Sites on mRNA by Fusing Multiple Sequence-Derived Features into a CatBoost-Based Classifier"

_ijms, 2023, doi:10.3390/ijms24097878_

Round 1

Reviewer 1 Report

ijms-2356536

The authors exploit a CatBoost-based method, m6Aminer, to identify the m6Am sites on mRNk, by nine different feature encoding schemes (Pseudo electron-Ion Interaction Potential, Hash decimal conversion method, Dinucleotide Binary Encoding, Nucleotide Chemical Properties, Pseudo k-tuple composition, Dinucleotide numerical mapping, K monomeric units, Series correlation Pseudo Trinucleotide Composition, K-spaced nucleotide pair frequency). Working by different assessment methods, 10-fold cross-validation, and independent test, The prediction model can be used to identify the m6Am sites in the whole transcriptome, laying a foundation for the function research of m6Am-  explaining the biological significance and application in the medical field for the progression of cancers.

Comments to the authors:

1. The introduction, materials and methods, and discussion part are well presented. 2. The math formulas and presented graphical analyses are at a high level. 3. The used references - only 13 (23%) are from the last 3 years. It needs to be moderate.

Minor editing of the English language required

Author Response

Reviewer 1:

General comments:

The authors exploit a CatBoost-based method, m6Aminer, to identify the m6Am sites on mRNA, by nine different feature encoding schemes (Pseudo electron-Ion Interaction Potential, Hash decimal conversion method, Dinucleotide Binary Encoding, Nucleotide Chemical Properties, Pseudo k-tuple composition, Dinucleotide numerical mapping, K monomeric units, Series correlation Pseudo Trinucleotide Composition, K-spaced nucleotide pair frequency). Working by different assessment methods, 10-fold cross-validation, and independent test. The prediction model can be used to identify the m6Am sites in the whole transcriptome, laying a foundation for the function research of m6Am-explaining the biological significance and application in the medical field for the progression of cancers.

Thank you, we are happy that you liked our paper.

Thank you so much for your feedback, we appreciate your efforts to improve our manuscript. Please see our response to each points below.

We are very grateful to the reviewer for their patience, careful reading of our manuscript and for providing us with excellent guidance to increase the accuracy of our results and enhance the paper's value. We have followed your advice and added a reply point-by-point. We will also be very grateful if you have further comments to enhance our paper.

  1. The introduction, materials and methods, and discussion part are well presented.

Thank you for carefully reading our manuscript and recognizing our work.  

  1. The math formulas and presented graphical analyses are at a high level.

Thank you for your high evaluation of our manuscript.

  1. The used references - only 13 (23%) are from the last 3 years. It needs to be moderate.

Thank you very much for this great comment. And kindly allow us to address this question. First of all, we totally agree with you that references from the last 3 years need to be moderate. We have increased the proportion of references to 21/52 from the last 3 years. Some other references related to mechanisms have been appropriately retained due to their significant position in this research field.

The reference “[39] Feng, C. Q.; Zhang, Z. Y.; Zhu, X. J.; Lin, Y.; Chen, W.; Tang, H.; Lin, H., iTerm-PseKNC: a sequence-based tool for predicting bacterial transcriptional terminators. Bioinformatics 2019, 35, 1469-1477.” has been replaced with “[39] Musleh, S.; Islam, M. T.; Qureshi, R.; Alajez, N. M.; Alam, T., MSLP: mRNA subcellular localization predictor based on machine learning techniques. BMC Bioinformatics 2023, 24, 109.”

The reference “[40] Yang, H.; Lv, H.; Ding, H.; Chen, W.; Lin, H., iRNA-2OM: A Sequence-Based Predictor for Identifying 2'-O-Methylation Sites in Homo sapiens. J Comput Biol 2018, 25, 1266-1277.” has been replaced with “[40] Fan, Y.; Wang, W.; Zhu, Q., iterb-PPse: Identification of transcriptional terminators in bacterial by incorporating nucleotide properties into PseKNC. PLoS One 2020, 15, e0228479.”

The reference “[41] Tang, Q.; Nie, F.; Kang, J.; Chen, W., mRNALocater: Enhance the prediction accuracy of eukaryotic mRNA subcellular localization by using model fusion strategy. Mol Ther 2021, 29, 2617-2623.” has been replaced with “[41] Feng, P.; Yang, H.; Ding, H.; Lin, H.; Chen, W.; Chou, K. C., iDNA6mA-PseKNC: Identifying DNA N6-methyladenosine sites by incorporating nucleotide physicochemical properties into PseKNC. Genomics 2019, 111, 96-102.”

The reference “[43] Lee, D.; Karchin, R.; Beer, M. A., Discriminative prediction of mammalian enhancers from DNA sequence. Genome Res 2011, 21, 2167-2180.” has been replaced with “[43] Orozco-Arias, S.; Candamil-Cortes, M. S.; Jaimes, P. A.; Pina, J. S.; Tabares-Soto, R.; Guyot, R.; Isaza, G., K-mer-based machine learning method to classify LTR-retrotransposons in plant genomes. PeerJ 2021, 9, e11456.”

The reference “[44] Liu, Z.; Dong, W.; Jiang, W.; He, Z., csDMA: an improved bioinformatics tool for identifying DNA 6 mA modifications via Chou's 5-step rule. Sci Rep 2019, 9, 13109.” has been replaced with “[44] Fletez-Brant, C.; Lee, D.; McCallion, A. S.; Beer, M. A., kmer-SVM: a web server for identifying predictive regulatory sequence features in genomic data sets. Nucleic Acids Res 2013, 41, W544-56.”

The reference “[46] Chen, Y. Z.; Tang, Y. R.; Sheng, Z. Y.; Zhang, Z., Prediction of mucin-type O-glycosylation sites in mammalian proteins using the composition of k-spaced amino acid pairs. BMC Bioinformatics 2008, 9, 101.” has been replaced with “[46] Zhao, Z.; Zhang, X.; Chen, F.; Fang, L.; Li, J., Accurate prediction of DNA N4-methylcytosine sites via boost-learning various types of sequence features. BMC Genomics 2020, 21, 627.”

The reference “[47] Wang, X.; Yan, R.; Song, J., DephosSite: a machine learning approach for discovering phosphotase-specific dephosphorylation sites. Sci Rep 2016, 6, 23510.” has been replaced with “[47] Chen, X.; Xiong, Y.; Liu, Y.; Chen, Y.; Bi, S.; Zhu, X., m5CPred-SVM: a novel method for predicting m5C sites of RNA. BMC Bioinformatics 2020, 21, 489.”

The reference “[49] Miao, F.; LI, Y.; Gao, C.; Wang, M.; Li, D., Diabetes prediction method based on CatBoost algorithm. Comput Syst Appl2019, 28, 215-218.” has been replaced with “[49] Kumar, P. S.; K, A. K.; Mohapatra, S.; Naik, B.; Nayak, J.; Mishra, M., CatBoost Ensemble Approach for Diabetes Risk Prediction at Early Stages. In 2021 1st Odisha International Conference on Electrical Power Engineering, Communication and Computing Technology(ODICON), 2021; pp 1-6.”

Due to changes in the references, some of the content in the article has also been fine-tuned accordingly (what added is marked blue).

  1. Lines 194 to 197

PseKNC has been successfully applied to the identification of RNA or DNA modification by forming physicochemical properties of oligonucleotides [38], such as MSLP [39], iterb-PPse [40] and mRNALocater [41].

  1. Lines 252 to 254

Ksnpf is an effective extraction method and has been successfully applied to the prediction of DNA N4-methylcytosine sites [46] and RNA 5-methylcytosine (m5C) sites [47].

Comments on the Quality of English Language: Minor editing of the English language required.

Thank you very much for your feedback. Regarding some issues with the English language in the manuscript, we have invited an expert with ten years of English teaching experience to polish our manuscript.

Reviewer 2 Report

It is of general importance to locate m6A sites. The m6Aminer program complements existing capabilities and improves such predictions by making them available online.

The manuscript is very well written and relevant literature is cited. The comparison of the predictions with the other two models, m6AmPred and DeepLm6Am, is convincing.

There are only a few issues:
1.) lines 73-75, This method selectively removes m6Am..........
Please make this description more understandable to non-specialists.
2.) Line 390, What does best stability mean for different datasets?
3.) The authors have developed a tool that competes with two other prediction methods. Therefore, the use of terms like "remarkably better" (line 100) or "better than" (line 426) should be avoided.
4.) In regard to 3.)The advantage of the method compared to others should possibly be explained in more detail in the discussion. Are there any examples for this?

5.) Figure 1. (A), etc
    2.2. feature
   , and so on (Line 38) imprecise formulations should be avoided

Author Response

Reviewer 2:

General comments:

It is of general importance to locate m6A sites. The m6Aminer program complements existing capabilities and improves such predictions by making them available online.

The manuscript is very well written and relevant literature is cited. The comparison of the predictions with the other two models, m6AmPred and DeepLm6Am, is convincing. Comments to the authors:

There are only a few issues:

Thank you very much for your important feedback and suggestions on our manuscript. Based on your valuable feedback on our manuscript, we will provide the following responses point by point.

  1. lines 73-75, This method selectively removes m6Am..........
    Please make this description more understandable to non-specialists.

Thank you very much for this valuable suggestion. This viewpoint is proposed by the reference we have cited. To make it easier for people to understand, we have expanded the content to a certain extent according to your requirements.

Lines 78 to 82

This method selectively removes m6Am while maintaining the integrity of m6A through in vitro demethylation reaction. When the identified substance cannot be eliminated by m6Am-seq, it is proven that the identified substance is m6A, otherwise m6Am. Therefore, this method can be used to distinguish m6Am from m6A at the 5 '- UTR.

  1. Line 390, What does best stability mean for different datasets?

Thank you very much for your comments, we have added the section reads as below.

Lines 400 to 403

Concurrently, the SD of six metrics for m6Aminer are lower than the other two models, which proves that m6Aminer has the best stability on different datasets. This best stability indicates that our model achieves stronger competitiveness in terms of generalization ability compared to the other two models in a completely new dataset.

  1. The authors have developed a tool that competes with two other prediction methods. Therefore, the use of terms like "remarkably better" (line 100) or "better than" (line 426) should be avoided.

Thank you, we agree with you, and have made the following changes to the content:

  1. Lines 108 to 110

Comprehensive comparison results show that our proposed model achieves competitive performance compared with the state-of-the-art predictors m6AmPred and DLm6Am.

  1. Lines 441 to 443

According to the comprehensive comparison results of this study, it could be seen that our predictor achieves competitive performance compared with m6AmPred and DLm6Am.

  1. In regard to 3.)The advantage of the method compared to others should possibly be explained in more detail in the discussion. Are there any examples for this?

Thank you very much for this great comment which highlights your experience in this field. And kindly allow us to address this question. We have supplemented this section in the main text.

Lines 437 to 441

To the best of our knowledge, the currently available models to predict m6Am using machine learning are m6AmPred and DLm6Am. Compared to the above models, m6Aminer has excavated more important features related to the m6Am site. What’ more, our model does not consume GPU resources and only takes litter training time.

  1. Figure 1. (A), etc 2.2. feature, and so on (Line 38) imprecise formulations should be avoided

Thank you for your suggestions for improving our manuscript. We have processed the unclear images and formulas in the text and uploaded high-resolution images separately in the revised version.

The issue of inaccurate formatting you mentioned has also been revised, and their specific positions in the text are on lines 118 and 131.
